# Optimising Seniors’ Metabolism of Medications and Avoiding Adverse Drug Events Using Data on How Metabolism by Their P450 Enzymes Varies with Ancestry and Drug–Drug and Drug–Drug–Gene Interactions

**DOI:** 10.3390/jpm10030084

**Published:** 2020-08-11

**Authors:** Roger E. Thomas

**Affiliations:** Department of Family Medicine, Cumming School of Medicine, University of Calgary, Calgary, AB T2M1M1, Canada; rthomas@ucalgary.ca; Tel.: +1-403-220-0157

**Keywords:** seniors, metabolism of medications, P450 cytochrome isoforms, genetic variability in metabolism of medications, drug–drug interactions, drug–drug–gene-interactions, DrugBank, Flockhart tables, Dutch pharmacogenetics working group, clinical pharmacogenetics implementation consortium

## Abstract

Many individuals ≥65 have multiple illnesses and polypharmacy. Primary care physicians prescribe >70% of their medications and renew specialists’ prescriptions. Seventy-five percent of all medications are metabolised by P450 cytochrome enzymes. This article provides unique detailed tables how to avoid adverse drug events and optimise prescribing based on two key databases. DrugBank is a detailed database of 13,000 medications and both the P450 and other complex pathways that metabolise them. The Flockhart Tables are detailed lists of the P450 enzymes and also include all the medications which inhibit or induce metabolism by P450 cytochrome enzymes, which can result in undertreatment, overtreatment, or potentially toxic levels. Humans have used medications for a few decades and these enzymes have not been subject to evolutionary pressure. Thus, there is enormous variation in enzymatic functioning and by ancestry. Differences for ancestry groups in genetic metabolism based on a worldwide meta-analysis are discussed and this article provides advice how to prescribe for individuals of different ancestry. Prescribing advice from two key organisations, the Dutch Pharmacogenetics Working Group and the Clinical Pharmacogenetics Implementation Consortium is summarised. Currently, detailed pharmacogenomic advice is only available in some specialist clinics in major hospitals. However, this article provides detailed pharmacogenomic advice for primary care and other physicians and also physicians working in rural and remote areas worldwide. Physicians could quickly search the tables for the medications they intend to prescribe.

## 1. Introduction

Primary care physicians prescribe medications for most illnesses and also renew medications prescribed by specialists. To optimise therapy and avoid adverse drug events (ADRs), it is important that physicians thoroughly understand the pharmacogenomics of all the medications they prescribe. Electronic medical records in community primary care do not provide pharmacogenomic advice and the prescribing advice in guidelines written by organisations of specialists is contained in many hundreds of very lengthy documents that are difficult to identify during the workflow of primary care physicians. For example, a panel of physicians aimed to reduce the number of cardiovascular guidelines and eventually made 89 key recommendations related to cardiovascular diseases from more than 400 existing recommendations from eight different guideline groups. [1]

Currently, primary care physicians select medications and dosages according to the guidelines they are able to locate by searching, then may need to wait to assess the effects of the medications on patients and receive the patient’s observations about possible ADRs. If the physician does not check the effects of prescriptions with blood levels and patients do not report problems, potentially inappropriate medications may be prescribed and patients experience undertreatment, overtreatment, or ADRs. The best solution is to provide primary care physicians with pharmacogenomic data and pharmacogenetic-based Decision Support Tools (DSTs).

## 2. Materials and Methods

### 2.1. Systematic Reviews

A search of Medline on 4 July 2020 using the search term pharmacogenomics identified 11,963 studies and 194,072 systematic reviews or meta-analyses, of which 221 were on pharmacogenomics. Of these 23 were of individuals ≥65 years. For Embase there 11,050 studies of pharmacogenomics and 349,392 systematic reviews or metanalyses of which 282 were on pharmacogenomics. Of these four were of individuals ≥65 years. Each systematic review was assessed for relevance and if authors did not provide a mean or median age, this was computed from all studies included in each systematic review. Evidence is reported in this article if available for those ≥65 or if the authors noted there were no differences in drug metabolism between those <65 or ≥65 years. Otherwise data from other relevant studies is presented until data are available specifically for those ≥65. A Medline search on 9 July 2020 identified 55,417 citations for electronic medical records and when combined with pharmacogenomics yielded 176 systematic reviews or meta-analyses. There were no summary studies of the overall costs for hardware, software, and training or performance across large healthcare systems and most publications were during 2014–2017 which reported initial experiences.

### 2.2. Pharmacogenetic Guidelines

The key sources of pharmacogenomic data are the DrugBank tables [2] and the Flockhart tables from the Department of Medicine at Indiana University [3] and they include the medications most likely to be prescribed by primary care physicians.

The pharmacogenomic data in the DrugBank tables are a comprehensive database of 13,580 drug entries including 2635 approved small-molecule drugs, 1378 approved biologics (proteins, peptides, vaccines, allergenics), 131 nutraceuticals, 6376 experimental drugs in the discovery phase and 5223 drug target/enzyme/transporter/carrier sequences. They are based on clinical trial reports, basic biochemistry research, and the detailed pharmacogenomic data that drug manufacturers are required to include in drug labels. DrugBank reports both the multiple P450 cytochromes which metabolise each medication and for those medications metabolised by complex enzymatic pathways with up to six pathways and each with from six to fifty enzymes [2]. The Flockhart tables were initiated by biochemist Dr. David Flockhart with his pharmacogenomic research on tamoxifen and are now maintained by the University of Indiana. The tables are clinically focused on the pharmacogenomic data for the nine main P450 cytochromes which metabolise 75% of medications. They also uniquely provide comprehensive data on which medications inhibit or induce single or multiple P450 cytochrome isoforms and which are strong or weak inhibitors or inducers. Inhibitors slow drug metabolism rates and cause higher drug levels and overtreatment and with prodrugs may result in potentially toxic blood levels. Inducers increase drug metabolism rates and cause lower drug levels [3]. RxFiles provides a convenient one-page list for primary care physicians. It is researched by a team of pharmacists at the University of Saskatchewan who also provide pharmacogenomic education to rural family physicians ([4], p. 211). These databases are not influenced by pharmaceutical companies.

In addition to the information in the Flockhart Tables and DrugBank two key groups, the Dutch Pharmacogenetics Working Group (DPWG) [5] and the Clinical Pharmacogenetics Implementation Consortium (CPIC) [6] provide detailed prescribing advice and cautions. These therapeutic pharmacogenomic guidelines identify which P450 genotypes affect the metabolism of individual drugs and provide advice about either avoiding medications or reducing dosages and also monitoring drug levels depending on the effect of the reduced- or enhanced-function alleles in each P450 cytochrome isoform. For both organisations, curators search the scientific evidence and the DPWG then uses quantitative methods to rate the level of evidence on a five point scale (0–4) and the clinical evidence on a seven point scale (AA-F) and the CPIC uses expert consensus to rate evidence on a three-point scale (weak, moderate, high).

### 2.3. Prescribing Using Curated Summaries of the CPIC and DPWG Pharmacogenomic Guidelines

The US Federal Drug Administration and the European Medicines Agency provide mandatory guidelines in their drug labels about indications and dosages. These guidelines are complex and not appropriate to the workflow of busy primary care physicians. For example, the CPIC guideline on CYP2C19 and clopidogrel is 45 pages, the CYP2C9 guideline on NSAIDS is 36 pages, the CYP2C9, VKORC1, and CYP4F2 and warfarin guideline is 8 pages and most other guidelines are 6–8 pages, with supplemental pages about additions. A study comparing their advice with the CPIC and DPWG guidelines noted differences and it is thus necessary to compare these sources [7].

A curated summary of the CPIC and DPWG guidelines provides succinct comparisons of the genotypes and the clinical advice for prescribing for each genotype is the quickest way for primary care physicians to access pharmacogenomic advice for prescribing. The latest version is by Bank [8] (Appendix A).

Bank’s summary of CPIC and DWPG advice reveals that the CPIC advice is more conservative than the DWPG. The following comments are, therefore, based on the more conservative advice of each guideline for medications commonly prescribed by primary care physicians.

An overview of the most serious concerns emphasised by the guidelines identifies the following medications. Warfarin is metabolised by CYP2C9 and CYP2D6 and its levels are also affected by the gene VKORC1 which affects Vitamin K metabolism so that there is strong advice to use a validated published pharmacogenetic algorithm that also incorporates patient data such as weight, age, and comorbidities. The summary for the pro-drug codeine notes the absolute contraindication in ultra-metabolisers (UMs) because of toxic and fatal levels of the metabolite morphine (an additional problem would be if patients increased the codeine dosage on their own initiative if they are not receiving the expected pain relief), and the subtherapeutic effect in poor metabolisers (PMs) (Appendix A).

Many medications are metabolised by multiple P450 isoforms. Five medications are metabolised by both CYP2C9 and CYP2D6 and for amitriptyline, clomipramine, doxepin, and imipramine the advice is to avoid for all patients except for CYP2C9 intermediate metabolisers (IMs) and then reduce the dose by 25% for CYP2D6 IMs. Because unknown interactions may occur with other medications these are drugs to be avoided. For clopidogrel, the advice is for all except UMs consider a drug not metabolised by CYP2C19. For phenytoin, a 25% dosage reduction should be considered for intermediate metabolisers (IMs) and 50% for PMs and drug levels should be monitored. Dosage reductions are advised for many medications and for simvastatin for patients with decreased function, alleles dosage should be reduced, an alternative statin considered, or drug levels monitored (Appendix A) [9,10,11,12,13,14,15,16,17,18,19,20,21,22,23].

### 2.4. Prescribing Using Curated Summaries of Drug–Drug and Drug–Drug–Gene Interactions

In a well-resourced large pharmacogenomic system with DST detailed advice will also be provided about drug–drug and drug–drug–gene interactions. With Next Generation Sequencing using long DNA reads even more detail will be provided including rare allele variants. This will require even more helpful DST advice to use this huge information increase. If primary care physicians do not have access to this advice, a curated summary by Bahar of drug–drug and drug–drug–gene interactions is very useful. It also adds more urgency to the advice in Bank’s [8,9,10,11,12,13,14,15,16,17,18,19,20,21,22,23] summary by providing numerical data for increases in medication levels with drug–drug and drugdelete–drug–gene interactions [24] (Appendix A, [25,26,27,28,29,30,31,32,33,34,35,36,37,38,39,40,41,42,43,44,45,46,47,48,49,50,51,52,53,54,55,56,57,58,59,60,61,62,63,64,65,66,67,68,69]) The two drug bases only partially overlap. Bank’s summary provides advice about nine medications (amitriptyline, citalopram, clomipramine, doxepin, escitalopram, imipramine, nortriptyline, phenytoin, and sertraline) [8] not in Bahar’s summary. Bahar’s summary [24] provides drug levels and advice about 43 drug–drug and drug–drug–gene interactions not in Bank’s table. The pharmacodynamics column of Bahar’s summary (Appendix A) is particularly important for the effects of specific patient genotypes. The most serious concerns highlighted are: NSAIDS taken with coumarins increase risk of anticoagulation (a 2.28 increased risk of INR ≥ 6 with alleles CYP2C9*2 or *3); for proton pump inhibitors with clopidogrel there is a 53% increase in inhibition of platelet aggregation; for diphenhydramine with metoprolol there are significant heart rate and systolic blood pressure reductions; for cotrimoxazole with venlafaxine severe tremors occur; for pantoprazole with atorvastatin myopathy and rhabdomyolysis are likely to occur, and for clarithromycin and codeine coma and respiratory depression are likely (Appendix A).

In the pharmacokinetics column of Appendix A there are numerous significant changes in medication levels for specific patient genotypes. The outcome measures reported for specific medications differ depending on the data source and Bahar [24] was not able to reduce them to a common metric.

A study of pain control in 187 patients (average age 58 years) after abdominal surgery illustrates the important effects of drug–drug–gene interactions. The patients were prescribed the pro-drug tramadol, a weak opioid which gives the most pain relief when transformed to its *O*-desmethyl enantiomer, and which has the highest affinity for μ-opioid receptors and thus intrinsic activity. The time-concentration area under the curve (AUC) for *O*-desmethyl tramadol varied markedly from 0 ng h/mL for PMs, 38.6 for NMs, 66.5 for EMs and 150 for UMs. In addition, those taking medications that inhibited CYP2D6 had significantly poorer pain relief than those taking no inhibitors and patients with poor pain relief asked for more medication [55].

There are many examples of drug–drug–gene interactions which result in very large increases in medication blood levels (Appendix A). For simvastatin taken with warfarin the advice is dose reductions of 25–43%; for fluconazole taken with flurbiprofen the area under the medication time-concentration curve (AUC) is increased from 41% up to 203%; for rifampin with tolbutamide clearance is increased from 70% up to 162%; for proton pump inhibitors (PPIs) with and clopidogrel the level of platelet inhibition is increased by 54%; for ticlopidine and omeprazole the AUC is increased from 401% up to 522%; for fluvoxamine and omeprazole the AUC is increased from 138% up to 462%; for fluvoxamine and lansoprazole the AUC is increased from 101% up to 1297%, for quinidine and venlafaxine the AUC is increased from 285% up to 1118%; for paroxetine and desipramine clearance is decreased from 20% to 78%; for paroxetine and aripiprazole the AUC is increased from 29% up to 136%; for diphenhydramine and metoprolol the AUC is increased from 61% to 90%; for celecoxib and metoprolol the AUC is increased from 36% up to 103%; for the CYP2D6 inhibitors paroxetine, amiodarone, cimetidine and ranitidine with tramadol the O-desmethyl tramadol AUC decreased by 82% to 93%. In the list of drug–drug–gene interactions the effect of clarithromycin on antifungals if CYP3A4 inhibitors are involved is specifically highlighted. Caution should generally be used with macrolides such as erythromycin and clarithromycin and also with anti-fungals because of the multiple isoforms that metabolise them (Appendix A) [24].

For illnesses likely to be treated by primary care physicians the most evidence is available for mental health and cardiovascular diseases and these will be focused on in the following sections, emphasising how although the pharmacogenomics is complex valuable decisions can be made for patients.

## 3. Results

### 3.1. Optimising Pharmacogenomic Prescribing for Illnesses Frequently Cared for by Primary Care Physicians

#### 3.1.1. Mental Illness

A major research focus has been using pharmacogenomics to improve prescribing for mental health. Seventy percent of all prescribing for mental health issues is undertaken by primary care physicians and they could focus first on the pharmacogenomics of prescribing in this key area or in cardiology to gain expertise in pharmacogenomics. Mental health consultations are usually of great salience for patients as they can affect functioning at work and in relationships and for physicians can often be difficult, lengthy, and require many repeated visits. If the patient either does not improve or worsens and also complains of side effects the patient and physician have an ongoing major problem. This can be avoided by applying pharmacogenomic knowledge.

#### 3.1.2. The Burden of Mental Illness in Primary Care

In the US the population has a 21% lifetime risk of depression and it is the sixth costliest disorder to treat [70]. An analysis of the 2006–2013 US Medicare Current Beneficiary Survey found that the average expenditure was $9177 but for depressed patients $14,436, adjusted for risk factors and comorbidities [71]. A study of the US Medical Expenditures Panel Survey 2004–2011 showed the major additional effect of comorbidities on the costs of depression. The average annual cost in 2010/11 for patients without depression or diabetes was $4818, with depression $8500, with diabetes $10,028 and with both depression and diabetes $16,518 [72].

#### 3.1.3. The Effectiveness of Usual Therapy for Depression

Most antidepressants block monoamine reuptake to increase monoaminergic neurotransmission, but the effectiveness of therapy is often reported by patients and physicians as suboptimal, with remission rates in clinical trials ranging from 30% to 40% [73].

Moreover, patients may have felt unwell during a substantial period of subthreshold depression and a systematic review of 16 studies (*n* = 67,318) of subthreshold depression found that individuals have a higher risk of developing major depression (IRR = 1.95; 95% CI 1.28–2.97) and this provides added pressure to find a solution when they present as clinically depressed [74]. Thus without pharmacogenomic guidance, there may be a substantial period of trial and error in trying medications to treat depression, exposing the patients to prolongation of symptoms, adverse relationship and work experiences, and suicidal ideation [75].

#### 3.1.4. How Pharmacogenomics Affects Response to Antidepressants

Genetic variants affect about 42–50% of the response rates of patients to antidepressants [76]. Within the Phase 1 enzyme P450 family two isoforms, CYP2C19 and CYP2D6 are the main pathways by which two thirds of all psychiatric medications to treat depression are metabolised. CYP2D6 is the most highly polymorphic of the isoforms resulting in substantial variability due to loss-of-function or gain of function in its 100+ alleles.

Therefore, it is necessary for primary care physicians to be aware of the most recent research on these two isoforms, their multiple genotypes, and how this variability can affect patients. The genotypes can include null alleles encoding non-functional proteins and patients are described as poor metabolisers, variants with decreased enzymatic or transcriptional levels resulting in intermediate metabolisers (CYP2C19*2, *3, and *4 are loss-of-function variant alleles), patients with two normal alleles (*1) are normal metabolisers, and individuals with the gain-of-function allele *17 or duplicated genes are ultra-rapid metabolisers. Null alleles reduce enzymatic activity more than the gain-of-function allele *17 increases activity. There are also a large number of rare individual allele variants that affect the metabolism of antidepressants. Additionally, many patients take multiple medications and these medications can act as either inducers or inhibitors of metabolism in these isoforms. A limited number of guidelines provide advice about the usual situation in which multiple medications are involved.

Because humans have been using pharmaceutical medications for only a few decades the enzymes that metabolise them have not been subject to selective evolutionary pressure. As a result, there is marked variability in alleles among humans. Each individual for each P450 isoform receives one allele from the mother and one from the father. It is important for primary care physicians to understand how the ancestry of each patient may affect the alleles they possess and how this affects the metabolism of all of their medications (Figure 1).

Westrhenen combined three large studies (total *n* = 4442 patients) of depressed patients and computed isoform enzymatic capacities which ranged from 0% for the Null/Null poor metaboliser phenotype to 150% for the Wtx3 ultra-rapid phenotype [78]. (Table 1).

For depressed patients their anti-depressants are metabolised principally by the CYP2D6 and CYP2C19 isoforms. It is necessary in prescribing to take account of the enzymatic capacities inherited by different ethnic groups for both isoforms. In CYP2D6 the normal function *1 allele is present in 33% of Europeans, 9% of Africans, 14% of East Asians, 26% of South Asians, and 40% of Americans. The missense *2 gene is a normally functioning allele and is present in 34%, 27%, 14%, 36%, and 33% respectively. The most frequent variants are the increased function allele *1XN (1%, 3%, 1%, 0.5% and 0.5%), and three decreased function alleles *10 (0.2%, 3.2%, 59%, 6.5%. and 0%); *17 (<0.1%, 20%, 0%, 0.1% and 0.7%), and *41 (3%, 3%, 3%, 14% and 3.5%).

For CYP2C19 the normal *1 allele is present in 59% of Europeans, 45% of Africans, 61% of East Asians, 52% of South Asians, and 77% of Americans, and the increased function allele *17 in 22%, 24%, 1.5%, 14% and 12%, and there is a large number of inactive allele variants (Figure 1) and ([77], pp. 693–695). Yet another complication is that apart from CYP2C19 and CYP2D6 in half of the genes that encode other enzymes that metabolise and transport anti-depressants rare variant alleles account for all their genetic variability, and these were not searched for in these studies so that the distributions of enzymatic capacities are likely to be even more varied [78].

One of the studies in Westrhenen’s group of studies was of 2087 patients (average age 48 years) in Oslo, Norway. The minimal therapeutic concentration of escitalopram was set as 25 nM, based on a serotonin reuptake (SERT) inhibition constant of 9.8 nM and a CSF concentration 1/3 that in serum [79]. Compared to the extensive metabolisers the escitalopram mean serum concentration was 3.3 times higher for the poor metabolisers and 1.6 times higher for the Null/*1 intermediate metabolisers but 20% lower for the *17/*17 ultra-rapid metabolisers [79]. Compared to the extensive metabolisers the poor metabolisers had a 31% rate (odds ratio 3.3) of switching antidepressant within a year despite more improvement in depression but because they experienced more adverse symptoms. The ultra-rapid metabolisers (*17/*17) also had a 29% rate (odds ratio 3.0) of switching medications but for a different reason, treatment failure due to low serum levels of antidepressant. Of the patients who switched, 34% switched to venlafaxine and 25% to another SSRI [79].

Another study in Westrhenen’s group of studies was of 1003 patients in Oslo, Norway. The ratio of the metabolism of venlafaxine in the primary metabolic pathway to O-desmethylvenlafaxine to N-desmethylvenlafaxine in the secondary metabolic pathway in the CYP2D6 isoform was used to identify variant alleles. The metabolic ratio varied from 0.47 for the Null/Null genotype to 68.18 for the *1/*4XN or *1/*41Xn ultrarapid metabolisers, demonstrating the importance of identifying the patients’ genomics to be aware of the wide ranges of metabolism [80].

Another meta-analysis of four studies (total *n* = 2558 patients, age 18–84) who were mostly receiving citalopram found the poor metabolisers had higher remission rates (OR 1.55, 95%CI 1.23–1.96; *p* = 0.00025) and although at 2 to 4 weeks they had more side effects (gastrointestinal, OR 1.26, 95%CI 1.08 = 1.47; *p* = 0.0033), (CNS, OR 1.28, 95% CI 1.07–1.53; *p* = 0.0068) and (sexual, OR 1.52, 95%CI 1.23–1.87; *p* = 0.001) there were no differences at 8–9 weeks. There were no differences in metabolic rates of antidepressants between those <65 and ≥65 years. Thus informing poor metabolisers that they would remit sooner than other patients and that the adverse effects would reduce to the same level as the other patients after a couple of months could be helpful to them in continuing their therapy [81].

#### 3.1.5. Prescribing Using the CPIC and DPWG Guidelines for Depression

Bank’s summary provides succinct comparisons of the CPIC [6] and DPWG [5] clinical advice for prescribing for each genotype and is the quickest way for primary care physicians to access pharmacogenomic advice for prescribing for depression [8] (Appendix A).

For the medications metabolised by both CYP2C9 and CYP2D6 (amitriptyline, clomipramine, doxepin, and imipramine) the advice is to avoid these for all patients except for CYP2C9 IMs and then reduce the dose by 25% if they are also CYP2D6 IMs. Considering unknown interactions may occur with other medications these are drugs to be avoided. The CPIC and DPWG guidelines provide strong advice about dosage reductions or choosing an alternative medication if patients have variant alleles in both P450 CYP2C19 and CYP2D6 that metabolise amitriptyline, clomipramine, doxepin and imipramine; variant CYP2C19 alleles that metabolise citalopram, escitalopram and sertraline; and variant CYP2D6 alleles that metabolise fluvoxamine, imipramine, nortriptyline and paroxetine.

The DrugBank [2] and Flockhart [3] tables indicate that these anti-depressants are also metabolised by and are inhibitors of multiple other CYP450 phenotypes, so that if other prescribed medications are competing to use the same phenotypes medication levels may not be as predicted by the CPIC [6] and CPWG [5] guidelines (Appendix A). Thus, prescribing should be guided by both the CPIC guidelines and the data in Appendix A in this article.

#### 3.1.6. Pharmacogenomic Decision Support Tools to Reduce Depressive Symptoms and Relapse Rates and Improve Remission Rates

The genotypes need to be identified by a laboratory test and are good for a lifetime. In 2020 thirteen panels of tests are available in Canada to assess patients’ pharmacogenomics relevant to medications for depression and 17% were associated with an actionable drug label or guideline for the patients tested. If family physicians could access such tests, obtain medication blood levels, use the guidelines, and consult with a pharmacist specialised in pharmacogenomics they could optimise their prescribing [82].

A systematic review assessed five randomised controlled trials (RCTs) of patients (total *n* = 877, average age 49 years) which used DSTs to use CYP2D6 and CYP2C19 genotype results to classify anti-depressant medications into three categories: green (use as directed), yellow (use with caution), or red (use with caution and also increase drug monitoring). The RCTs were at low risk for randomisation and attrition on the Cochrane Risk of Bias Tool but all were at high risk for industry sponsorship, recruitment of patients, and absence of blinding of the treating physician, and there was significant heterogeneity in effect sizes (I^2^ = 71%). Remission rates (measured by a score ≤7 on the 17 item Hamilton Depression Rating Scale) were higher in the group which received pharmacogenomic prescribing advice (RR 1.71; 95%CI 1.17–2.48) [83].

Another RCT randomised 1541 patients ≥18 years (with at least one failed trial of antidepressant treatment to either an arm in which “combinatorial pharmacogenomic test results were available to guide treatment decisions” (as directed, caution, or increased caution and with more monitoring) or to treatment as usual. Symptom improvement was defined as the percent change in the HDRS-17 score, remission as a score ≤7, and response as an improvement of 50% in the score. Average symptoms improved with a decline in HDRS-17 scores from baseline to eight weeks by 27% in the guided-care arm and by 22% in the treatment-as-usual arm; the response rates were 27% and 19%, and the remission rates 18% and 11%. A sub-sample of 912 patients (142 ≥ 65 years) had an average of 3.6 failed medications and an average HDRS-17 score of 20. These participants were shown later by genetic testing to have had gene-drug interactions for their initial medications and when they changed to medications assessed later by genetic testing as appropriate at the 24-week follow-up in the guided therapy arm there was a 42% decrease in scores, a 44% response rate, and a 33% remission rate compared to baseline [84]. There was no analysis whether there were different outcomes for those ≥65.

#### 3.1.7. Cost Effectiveness of Pharmacogenomic-Guided Therapy for Depression

A doubled blinded RCT of depressed patients who had failed at least one medication on a combinatorial pharmacogenetic test panel compared costs for their next prescriptions which were guided by pharmacogenomics. For 269 patients ≥65 when cared for by a primary care physician for 26 CNS, neurology and psychotherapeutic drugs the annual costs were US$ 4526 when the medications were incongruent with the pharmacogenomic test recommendations and US$ 2789 when congruent (savings $1737) and for those <65 $5521 and $3196 (savings $2325). The patients had multiple medical conditions and when the costs for all their medications in addition to the central nervous system medications were computed for those ≥65 they were US$ 11,088 when the medications were incongruent with the pharmacogenomic test recommendations and US$ 6974 when congruent (savings $4113) and for those <65 $10,311 and $6439 with savings of $3872) [85].

## 4. Translating Pharmacogenomics into Effective Clinical Practice for Primary Care Patients with Other Than Mental Health Problems

Pharmacogenomic research for medications prescribed by primary care physicians in areas other than mental health will now be commented on.

### 4.1. Studies of Pharmacogenomic-Guided Therapy for Multiple Illnesses

A study of 5,429,266 US Medicare patients ≥65 years 2009–2012 found that 2.4% were taking drugs classified by CPIC guidelines as at the highest priority for adverse events and 4.4% classified by DPWG as being in significance classes C to F (C = long standing discomfort without permanent injury attributed to the medication; D = permanent symptom or invalidating injury; E = failure of life-saving therapy; and FB = death, arrhythmia or unanticipated myelosuppression) [86]. Thus there is a burden of ADRs which family physicians are ideally positioned to treat because they have a wide scope of practice.

An RCT randomised 342 patients (average age 75) to a DST with pharmacogenomic results (PGx), or to DST without PGx, or control. PGx testing was planned for cytochromes CYP2D6, CYP2C19, CYP2C9, CYP3A4, CYP3A5, and VKORC1. The patients randomised to the DST support with PGx group received a phone call informing them about the trial, were mailed a buccal sample kit, and requested to mail the sample to the lab. However, this procedure proved to be a barrier and only 43% mailed in a sample; those who did not provide a sample were then added to the DST without PGx support arm. Pharmacists blinded to the arm assignment identified drug therapy problems (DTP) and ranked them from Grade 1 (mild symptoms), Grade 2 (moderate, minimal intervention required), Grade 3 (severe or medically significant but not immediately life-threatening, hospitalisation required), Grade 4 (life-threatening, urgent intervention), to Grade 5 (death related to ADRs). The pharmacists made a total of 1054 (average 3.08 per patient) DTP recommendations and the more serious recommendations were more likely to be accepted by the prescribers (OR 1.96, 95%CI 1.00–3.84) [87].

An observational cohort study matched 205 patients ≥65 tested with the DST *YouScript* which predicts drug levels, to a propensity-matched control group of 820 untested patients from the MORE Registry database of 9.7 billion US medical events. Specialised pharmacists then made recommendations to prescribers. For the 294 hospitalisation, emergency or outpatient visits of the tested patients during the four months follow-up the mean difference in costs per tested patient compared to untested was US$1132 for an annual cost difference of US$4528 (net US$3614 after deducting the US$914 costs of testing). The tests would be valid for a lifetime if testing was comprehensive [88].

### 4.2. Pharmacogenomic Studies of Cardiovascular Medication Dosing Requirements

Warfarin still remains the most frequently prescribed anti-coagulant worldwide, and family physicians need to be thoroughly conversant with the complicated pharmacogenomics.

#### 4.2.1. Warfarin and P450 CYP2C9

Primary care physicians can make a major contribution to the care of their anti-coagulated patients by ensuring they have a genotype assessment. There are multiple indications for warfarin: atrial fibrillation, mechanical heart valves, deep venous thrombosis, and pulmonary embolism. Warfarin prevents γ-carboxylation of the vitamin K-dependent coagulation factors II (prothrombin), VII, IX, and X. Patients on warfarin are within therapeutic range on average only 2/3 of the time and it is necessary to identify individuals with CYP2C9 variant loss-of-function alleles who need lower doses of warfarin. A variant in CYP4F2 involved in vitamin K metabolism is also a risk factor for bleeding associated with warfarin. The opposite problem, warfarin resistance, is due to loss-of-function variant alleles in VKORC1 which control the vitamin K epoxide reductase complex, resulting in higher warfarin dosages needed to achieve therapeutic anticoagulation levels. There are ancestry groups with multiple variant alleles, and one example is Ashkenazi Jewish populations who tend to need higher warfarin dosages [89].

The Genetic Informatics Trial of 1650 patients (average age 72 years) who had knee or hip surgery assessed four adverse endpoints (INR > 4, major bleeding within 30 days post-surgery, venous thromboembolism within 60 days post-surgery, or death). In 808 patients receiving genotype-guided warfarin dosing 87 (10.8%) had at least one adverse endpoint and in the clinically guided group 116 (14.7%), (RR = 0.73, 95%CI 0.56–0.95). However, 65% of all endpoints were an INR ≥ 4 (56 in the genotype-guided group vs. 77 in the clinically guided group) (RR, 0.71; 95% CI, 0.51–0.99) and no other endpoint reached significance. In the genotype-guided group there were two episodes of major bleeding (eight in the clinically guided group), 33 venous thromboembolisms (38 in the clinically guided), and no deaths. There was a small improvement in the percentage of time in the therapeutic range for genotype-guided (54.7%) compared to clinically guided (51.3%) dosing [90,91].

For warfarin, the CPIC guideline [6] lists the variant alleles for CYP2C9 and gives strong advice to use a validated published pharmacogenomic algorithm to compute dosage. The DPWG guideline advice [5] differs and advises for allele *1/*2 Initiate therapy with the recommended starting dose [4A = minor clinical effect], for 1/*3 [4D = risk of permanent symptom or invalidating injury] and *2/*2 [4A] consider a reduction to 65% of the normal starting dose, and for both *2/*3 [4A] and *3/*3 [4C = clinical effect of longstanding discomfort (48–168 h) without permanent injury] consider a reduction to 45% of the normal starting dose. (Appendix A). It would clearly be wise to use an algorithm that incorporates key data like age, weight, and comorbidities, as well as other medications to compute recommended doses.

#### 4.2.2. Warfarin and VKORC1

There are four VKORC1 genes that affect warfarin dosing levels. A meta-analysis of 53 studies found that compared to patients with the normal metaboliser type (-1639 AA), patients with VKORC1 -1639 types GA, GG, and G had daily average higher warfarin maintenance dosages of 45%, 77%, and 51% respectively. Compared to patients with type 1173 TT, those with type 1173 CC, TC, and C, had daily average higher warfarin maintenance dosages of 83%, 26%, and 53%. Compared to patients with type 3730 GG, those with type 3730 AA, GA, and A had daily average higher warfarin maintenance dosages of 40%, 25% and 33% respectively. There are also important differences by age and ancestry. The VKORC1 -1699AA homozygote is carried by 82% of individuals of Japanese, 82% of Chinese and 14% of European ancestry.

In the ENGAGE AF-TIMI 48 trial of 14,348 patients age ≥ 21 with atrial fibrillation and who had received a genetic analysis, 4833 were assigned to warfarin therapy. The study shows clearly in a 7 × 4 table how the possible combinations of allele variations in CYP2C9 and VKORC1 result in three bleeding risk groups (Figure 1 in ref [92]). There were no differences by age between the genotypes. If the patient has abnormal gene VKORC1 alleles the CPIC and DRWG advice differs. The CPIC for phenotypes -1639GA and -1639AA provides strong advice to calculate the dosage based on a published validated pharmacogenetic algorithm [6]. The DPWG advice for phenotype -1639GA is to initiate therapy with the recommended starting dose and for phenotype -1639AA consider a reduction of 60% in the recommended starting dose [both 4A advice] [5] (Appendix A). Again, an up-to-date algorithm that allows for weight, age, comorbidities and other medications is strongly indicated.

#### 4.2.3. New Oral Anticoagulants (NOACs)

Because of the problems of subtherapeutic dosing or bleeding associated with warfarin many patients have changed to NOACs. The ENGAGE AF-TIMI 48 trial compared both the NOAC edoxaban to warfarin and also the effects of different CYP2C9 and VKORC1 alleles on warfarin levels. The trial randomised patients ≥21 years with non-valvular atrial fibrillation within the previous 12 months and a CHADS2 risk score ≥2 to either warfarin, or edoxaban 30 mg, or 60 mg daily. Edoxaban doses were reduced by half at randomisation or during the trial based on bodyweight, renal function, and use of P-glycoprotein inhibitors [92]. The FDA bleeding risk designations were used to classify different types of patient response to warfarin. Normal responders to warfarin (62%) were designated as those with CYP2C9*1/*1 and VKORCI G/G or G/A, or CYP2C9*1/*2 and G/G; highly sensitive (2.9%) as those with CYP2C9*3/*3 and VKORC1 G/G, or G/A or A/A, or *2/*3 and G/A or A/A, or *2/*2 and A/A, or *1/*3 and AA; and all the remaining combinations as sensitive responders (35%) [92]. During the first 28 days post-operation the 2982 normal responders spent 32.5% of their time with an INR of 2 to 3, 48.9% with an INR < 2 and 9.5% with an INR > 3; the 1711 sensitive responders 31%, 32%, and 24%, and the highly sensitive 42%, 29%, and 31%. Within the first 90 days 31 (1%) of the normal responders had a major bleed and 109 (3.8%) a clinically relevant non-major bleed; sensitive responders 23 (1.4%) and 78 (4.7%) and the highly sensitive 3 (2.3%) and 18 (13.4%). Compared to warfarin both edoxaban 30 mg/day (OR 0.70 95%CI 0.64–0.75) and edoxaban 60 mg/day 0.90 (0.83–0.97) had a lower risk of bleeding. Both edoxaban dosages were non-inferior to warfarin for prevention of stroke and systemic embolism and had a significantly lower rate of cardiovascular death [92].

#### 4.2.4. Direct Oral Anticoagulants (DOACs)

There are currently three Direct Oral Anticoagulants (DOACs): apixaban, dabigatran, and edoxaban. A warfarin overdose can be corrected by an injection of vitamin K. In patients experiencing a major bleed on rivaroxaban or apixaban reversal of the inhibition of factor Xa by can be provided by Andexanet alpha, a recombinant modified human factor Xa decoy protein group. After use of Andaxanet alpha by 12 h later 79% had effective hemostasis but during the 30-day follow-up 12/67 (18%) had a thrombotic event showing the need for consideration of prompt reinstatement of therapy [93,94,95].

Thus pharmacogenomics demonstrate clearly the care that is needed when prescribing warfarin, that even normal responders spend 58% of their time in the first 28 days outside the 2 to 3 INR range, and that NOACS and DOACs overall have clear benefits [92].

#### 4.2.5. Clopidogrel

For clopidogrel, the CYP2C19*2 loss-of-function variant results in lower clopidogrel levels causing increases in major adverse cardiovascular events and also in adverse events after cardiac catheterisation [96,97,98].

Of 651 patients (average age 70) who experienced either a myocardial infarction (*n* = 555, 85%) or an ischemic stroke (*n* = 91, 14%) and were started on clopidogrel in the following 24 months 299 (46%) had either another arterio-occlusive event or died. Those with the CYP2C19*2 loss-of-function allele had an increased hazard ratio of 1.29 (1.04–1.59; *p* = 0.019) compared to those who were not carriers [99].

Loss-of-function alleles are also important in rendering clopidogrel less effective in preventing stroke recurrence. A study of 2933 Chinese patients with a transient ischemic attack or minor stroke found that the 42% who had loss-of-function variants CYP2C19 *2 or *3 had lower protection from recurrent TIAs or strokes [100].

The importance of pharmacogenomic guidance is shown by a systematic review of five studies (total *n* = 2900 patients, average age 67 years) of the effects of CYP2C19 variants on the outcomes of myocardial infarcts, revascularisation, stroke or death. For patients who received genotype-guided therapy 125 (8.9%) had one of these outcomes and in the conventionally dosed patients 239 (15.9%) RR 0.54 (95%CI 0.41–0.72; I^2^ 30%) [101].

The CPIC and DRWG advice for clopidogrel is the same but the advice is given with different strength levels. For clopidogrel the CPIC advice for IM and PM patients is to consider an alternative drug not metabolised by CYP2C19, and for UM patients initiate therapy with the recommended starting dose [6] The DPWG for IM and PM the advice is use an alternative drug not metabolised by CYP2C19 (the advice is level F = risk of death or other serious outcome) and for UM initiate therapy with the recommended starting dose [5] (Appendix A).

#### 4.2.6. Sulfonylureas and Cardiac Arrhythmias

A pharmacogenomic genome-wide association study of eleven cohorts of individuals mostly 65 or older (*n* = 71,857) of European, African-American and Hispanic/Latino ancestry assessed the use of sulfonylureas for diabetes mellitus and changes in electrocardiogram (ECG) measurements of three ECG phenotypes (QT, JT and QRS intervals). Eight novel pharmacogenomic loci met the threshold for genome-wide significance (*p* = 5 × 10^−8^) with prolongation of the intervals by >5 ms, which is the US Federal Drug Administration’s threshold for regulatory concern [102].

### 4.3. Tamoxifen and Breast Cancer

Patients are likely to ask their primary care physician to renew their tamoxifen prescription, ask whether it is effective and should they continue it. It is important to check if the patient’s tamoxifen blood levels have been measured, and if they are low if she has had a pharmacogenomic investigation. It is also important to verify in planning therapy that blood leucocyte or saliva samples were obtained to identify SLCO1B1 alleles because tumour cells may lose these alleles and allele studies based on tumour cells or formalin-fixed paraffin-embedded samples may demonstrate fewer alleles [103].

More than 50% of breast cancer patients are estrogen receptor-positive. Tamoxifen binds to and causes a conformational change in estrogen binding to these receptors and alters the transcription of estrogen-dependent genes, resulting in a decrease in breast cancer relapse rates of 40% [103]. P450 CYP2D6 metabolises tamoxifen to N-desmethyl-tamoxifen and then 92% of this metabolite becomes endoxifen and 8% becomes 4-H Tamoxifen. Both metabolites have a 100-fold higher affinity for estrogen receptors and 30- to 100-fold greater suppression of estrogen-dependent cell proliferation [104,105].

CYP2D6 is highly polymorphic and has >100 functional alleles, including whole gene duplications and deletions. A meta-analysis of 4973 breast cancer patients (average age 65) from twelve globally distributed sites found there was no relationship between cytochrome P450 CYP2D6 variant allele status and clinical outcomes with tamoxifen therapy, but across sites there was strong heterogeneity of results. The authors in a separate study described how they then harmonised data across the 12 sites and defined post-hoc a subgroup of 1996 postmenopausal women who were estrogen receptor-positive and had received tamoxifen 20 mg/day for five years. For these women there was a statistically significant effect of CYP2D6 poor metaboliser status on worsening both invasive disease-free survival and the breast cancer-free interval with little heterogeneity across sites (hazard ratio 1.25, 95% CI 1.06–1.47, *p* = 0.009). This indicated a need to change therapy for the PM group [104,105].

For women with low endoxifen levels toremifene may be a suitable alternative. A Cochrane meta-analysis of seven RCTs of patients (average age 53) with advanced breast cancer found that tamoxifen and toremifene are equally effective [106]. A study of poor tamoxifen metabolisers investigated whether switching them to toremifene would provide better levels because the chlorine atom on the side chain of toremifene prevents 4-hydroxylation by CYP2D6. For 182 patients taking tamoxifen 20 mg/day average serum endoxifen levels were 23.5 ng/mL with a wide range from 1.8 to 64.5 ng/mL. For the 61 patients taking toremifene 40 mg/day for the two metabolites the average concentration of the 4OH metabolite was 8.1 ng/mL and for the 4OH-NDM metabolite was 15.4 ng/mL. For the 30 patients taking toremifine 120 mg/day the average concentrations of the two metabolites were 16.0 and 32.6 ng/mL, a marked improvement [107].

For women experiencing depression due to the cancer and hot flashes due to tamoxifen it is important not to prescribe SSRIs and SNRIs, which are moderate to strong CYP2D6 inhibitors and decrease endoxifen levels by 50–72% [103].

### 4.4. Benzodiazepine Use and Pharmacogenomic-Related Increased Risk of Falls

A summary of three prospective studies of falls included 11,485 community-dwelling individuals (average age 72) in the Netherlands (the Rotterdam, B-PROOF, and LASA studies) of whom 3705 (32%) had a recorded fall during 91,996 follow-up years. The percentages prescribed benzodiazepines varied across the studies from 4% to 15%. Compared to nonusers of benzodiazepines, users who were CYP2C9*1 homozygotes (normal metabolisers) had no increased fall risk (HR 1.14; 95% CI 0.90–1.45) but the fall risk was higher in three groups with abnormal alleles. Those with a CYP2C9*2 or *3 allele had a 45% increased fall risk with an HR of 1.45 (95% CI 1.21–1.73), heterozygous allele carriers a 30% increased fall risk with an HR of 1.30 (95% CI 1.05–1.61) and homozygous allele carriers a 91% increased fall risk with an HR of 1.91 (95% CI., 1.23–2.96). Falls are an important cause of morbidity and mortality for seniors and thus this information can be used to help encourage seniors to deprescribe their benzodiazepines [108].

### 4.5. Statins and Myopathy

Statins are metabolised by gene 1B1 SLCO1B1A which is a solute carrier organic anion transporter family member and has two important variants. A meta-analysis of 14 studies with 3265 patients with myopathy compared to 7743 controls without myopathy found that the risk of myopathy in patients taking a statin was increased if they had gene variants SLCO1B1 521CC (OR 2.31, 95% CI 1.15–4.63, *p* = 0.019) or 521TC (OR 1.34, 95% CI 1.02–1.76, *p* = 0.034) or the combination of 521CC and 5221TC (OR 1.82, 95% CI 1.32–2.51, *p* < 0.001). Only the combination of CT + TC alleles showed any difference in the odds ratio of myopathy by age: <65 (2.04; 1.53–2.71), >65 (1.31; 0.77–2.21) (*p* = 0.003) [109].

In a systematic review of four RCTs and 12 non-randomised studies of patients taking high dose simvastatin therapy of 80 mg/day for the 85 participants with myopathy their odds ratio for myopathy was 4.5 (95% CI 2.6–7.7) per copy of the C allele, and 16.9 (95% CI 4.7–61.1) for those with two alleles (CC) compared to the normal TT homozygotes [110]. The PharmGKB database assesses the evidence as level A (highest level evidence) that statins cause myopathy in patients with gene SLCO1B1 allele variants [111].

For simvastatin the CPIC and DPWG guidelines differ. The CPIC advises for both of the decreased function alleles 521TC and 521CC to prescribe a lower dose or consider an alternative statin such as pravastatin or rosuvastatin and consider routine surveillance of creatine kinase levels [6]. The DPWG guideline for 521TC advises consider an alternative drug or if simvastatin is warranted prescribe a maximum of 40 mg daily, and for 521CC select an alternative drug [5] (Appendix A).

### 4.6. Genetic Variations in Uric Acid Levels with Allopurinol Treatment, and Adverse Skin Reactions to Allopurinol

The Kaiser Permanente Genetic Epidemiology Research on Adult Health and Aging (GERA) cohort is a sample of 110,266 adult members of the Health Maintenance Organisation who have received a genetic assessment including high-density single nucleotide polymorphism (SNP) markers. For a subsample of 2027 patients (average age 68) with gout, the effects of SNP genetic variations on changes in uric acid level in patients taking allopurinol were assessed. The gene ABCG2 encodes the efflux pump BCRP and was associated with a reduction in uric acid levels in non-Hispanic white patients (*p* = 2 × 10^−8^), but a missense allele (rs2231142) was associated with a reduced response to allopurinol (*p* = 3 × 10^−7^) [112].

Allopurinol to inhibit gout attacks can cause skin reactions. A meta-analysis of 21 pharmacogenetic studies of 2,921 patients taking allopurinol included 551 patients with skin reactions and 2370 without reactions and compared them to 9592 healthy patients without skin rashes. Carriers of HLA-B*58:01 had a greatly increased odds ratio for skin reactions of 82.77 (95% CI 41.63–164.58, *p* < 10^5^). Therefore, in new gout patients, it is prudent to check HLA-B carrier status [113].

The next section describes how participating in a large pharmacogenomic care system is especially helpful to primary care physicians.

## 5. Becoming Part of a Large Pharmacogenomic Prescribing System

The US National Human Genome Research Project is one of the key funders of genomic research worldwide. A search of their website did not identify any reports on the overall costs and achievements of genomic research although there are reports on highly focused research projects. [114]. The next Research plan will be published in October 2020 [115].

The total number of patients who have received pharmacogenetic testing worldwide has not been published, is known only by the testing companies and varies by country and local healthcare systems. The larger insurers in the US are increasingly testing for specific indications such as when patients fail to benefit from antidepressants. A project of the provincially owned Alberta Precision Laboratories is to standardise these tests. It is cheaper to obtain whole genome tests than for individual genomes [116]. A review of tests relevant to psychiatric patients available in Canada as of 30 May 2019 identified 13 tests available in all provinces but only two had peer-reviewed RCT tests that supported their use. The costs ranged from CA$199 to CA$2310 with a median of CA$499. For five tests the results could be reported directly to patients [82].

Currently, preliminary research is underway to test the effect of reiterative replacement of medications in the lists of patients with polypharmacy to reduce the potential number of drug interactions [117].

This article so far has described how a primary care physician can improve prescribing by using pharmacogenomic knowledge. To join a large well-financed and well-resourced system confers major benefits in prescribing by obtaining access to cheaper genome-wide tests which have undergone standardisation assessments in expert laboratories and also advice available through large systems.

### 5.1. Improving EMR Medication Recording

Increasing EMR accuracy involves mandating responsible persons to update and maintain the accuracy of patients’ medication lists, including medications stopped and added by emergency departments, hospitals and consultants. This also includes non-prescription medications, naturopathic medications, and herbals (although literature searches show that there is little knowledge about their pharmacogenomics).

The ideal situation is in Holland where the patient is typically listed with one GP and one pharmacy and both maintain the patients’ EMRs. A study of pharmacist-initiated pharmacogenetic testing of 200 patients found that pharmacists recorded 96% and GPs 68% of the phenotypes and over a two-and-a-half year period 24% of the medications prescribed had actionable drug–gene interactions [118]. In some countries, patients may be unsure who is responsible for updating their medication list. In a large region in Germany a survey of 5340 citizens ≥ 65 years (4% of the region’s population) were asked about medication lists and of the 49% who responded 74% said their general practitioner was involved in issuing their medication list and 9% that the patient was solely responsible for the list. Importantly, only half of the lists had been updated in the previous year [119].

At the health system level if records are not being updated promptly by prescribing physicians and dispensing pharmacists working as a team this should be identified, help provided and coordinated within the health system for all organisations providing care such as hospitals. The health system also needs to tell patients that they are expected to bring all their medications including non-prescription medications to each visit to any health organisation to verify changes and avoid errors, and thus coordinate records immediately across all care providers. Otherwise, inaccurate medication lists will result in inaccurate computer decision support.

### 5.2. Ensuring DST Systems Provide Comprehensive Advice Valued by Both Patients and Physicians as Improving Care

Health systems need to provide skilled resources and finance to ensure their EMR provides pharmacogenomic DSTs to avoid adverse drug reactions and drug allergies so that it becomes a valuable tool that physicians regularly find essential and does not cause them to have alert fatigue.

Major funding has been provided in many high-income countries to improve EMR use, but often has not focused on effects on patient-physician relationships or providing readily usable pharmacogenomic advice. The American Recovery and Reinvestment Act directed $27 billion for Medicare incentive programs to help office-based physicians implement EMRs and payments depended on EMR implementation, demonstration of meaningful technology use, and attestation processes and certification but provision of pharmacogenomic advice was not included [120].

Time on EMRs can discourage physicians from providing detailed pharmacogenomic advice. A study of 142 family physicians in a system in southern Wisconsin over three years found that on average during their 11.4 h workdays they spent 5.9 h in the EHR each weekday (4.5 h during clinic and 1.4 h after clinic hours). Clerical and administrative tasks (documentation, order entry, billing, coding, and system security) took 44% and managing their inboxes took 24% of their total time in the EMR [121]. The EMR can also detract from patient-physician relationships. A study of eye-gaze patterns during 100 video-recorded patient visits showed that when using the EMR compared to paper charts physicians spent less time looking at patients and patients also almost always looked at the computer screen whether they understood what was on the screen or not [122].

Therefore, EMRs should display information about medications, their interactions and ADRs so that patients can easily understand and as a team discuss them with their physician. The medication list, pharmacogenomics, and prescribing advice in the EMR must be simple and easily shareable with patients so that both physicians and patients assess that its use adds greatly to empathetic care and listening, aspects of care valued by patients.

### 5.3. Progress in Implementing Pharmacogenetic Guidelines in Large Health Care Systems

Because of the technical and financial complexity of pharmacogenomics, implementation has been mainly in large university and health care systems and associated clinics. The US National Human Genome Research Institute’s (NHGRI) Implementing GeNomics In pracTicE (IGNITE) Network [123] funds six major university medical institutions and 14 community partners to implement genomic medicine. The member institutions identify several key factors ensuring sustainability: health care worker education, development of clinical genetic decision tools for medications which may have multiple drug–drug–gene interactions and be prescribed by a wide range of specialties, high quality cost effectiveness studies to justify use and reimbursement, impacts on health care workers’ workflows rated as beneficial, and reimbursement for genetic testing [124,125,126].

The NHGRI also funds the eMERGE Network of three large health care systems, three pediatric medical centres and four academic medical centres. They also all prioritised development of pre-emptive algorithms and point-of-care resources linked to CDS alerts. Physicians, genetic counselors, pharmacists and nurses reported unfamiliarity with genetic testing, a lack of confidence implementing pharmacogenomics, and insufficient time for education. Implementing and evaluating education was not funded, and centres independently developed Clinical Decision Support Knowledgebase educational materials [127] and three sites developed formal training materials. At the Mayo Clinic the physician faculty development programme for physicians commencing their careers focused on using pharmacogenomics in practice, and a grant supported a team to develop and coordinate education across the system and sought respected exemplars to champion the use of pharmacogenomics. At other sites, problems were the large workload in developing peer-reviewed clinical education materials and delivering them across large institutions, and difficulties providing continuity in education because of the high mobility of learners in large institutions [128,129]. Thus, for primary care physicians to practice even some elements of pharmacogenomics requires excellent financial and technical support.

### 5.4. Developing Team Work to Deliver Pharmacogenetic Advice

There is a major need to educate health care personnel to understand and clinically implement genetic test results, and for physicians, pharmacists and nurses to work together as teams to discuss medication appropriateness and their mutual responsibilities and liabilities. Ongoing evaluation of guideline utility is necessary, and the SPARK toolbox has been developed for this purpose [130].

### 5.5. Integrating Pharmacogenomic Data into Electronic Health Records

EMRs have usually developed without provision for genomics [131] and there is a need for developers and vendors to integrate genetic results in an optimum location in the EMR. Genetic tests will be accurate for a lifetime, results may guide current and future therapies and predict future adverse drug reactions and allergies. They thus need to be in a central place in the EMR to be found promptly by all other providers and move with patients’ records if they change location. The ideal location for pharmacogenomics results would be the problem list cross-referenced to the medications and allergies lists. However, Genotypes and Phenotypes have not yet qualified as ICD-10 codes. Physicians may find it difficult to enter pharmacogenomics phenotypes, so a central list of actionable genotype-phenotype results needs to be easily accessible to be copied into the EMR and which will trigger CDS interventions. The correct entry of new pharmacogenomic data and use of CDS solutions needs to be accomplished with a few key strokes in the EMR. Pharmacogenomic results need to usefully and briefly summarise data and not be filed in long difficult-to-find pdfs. The Mayo clinic has neatly summarised the task:

“Because these pop-up alerts interrupt the clinician workflow, they should have a carefully crafted user interface to provide a clear reason for the alert, the level of the alert’s importance, explicit recommendations for clinical treatment, and easy access to the recommended action (ordering a test, changing a medication, changing the dose of the medication, documenting an exception, etc.). Due to the limited space in the pop-up alert, links to additional education material online may be helpful.” ([132], p. 257).

### 5.6. Decision Support Tools

DSTs should provide accurate up-to-date clinical and genetic information and be welcomed as simplifying workflows. The time taken during the patient visit should be assessed as very valuable in improving medication decisions and preventing serious and avoidable mistakes. Tools have been developed to evaluate the usefulness of DST [133].

To reduce alert fatigue interruptive DST alerts need to make pharmacogenomic data optimally relevant by linking with patient data. An example is the higher risk of patients with the HLA-B*15:02 variant allele experiencing the serious allergic reactions of Stevens-Johnson syndrome and toxic epidermal necrolysis. Patients of Asian ancestry are more likely to have this allele and all health care workers who enter data in the EMR need to be educated about these risk relationships and be meticulous in documenting patient ancestry [134].

The Mayo clinic identified the complexity of providing simple but sophisticated DST tools to solve pharmacogenomic problems:

“CDS interventions that alert clinicians of potential drug–gene interactions when they are ordering a medication are one of the most important safety nets to prevent the occurrence of PGx-related ADRs. However, these pop-up alerts require up-to-date structured PGx data to be accessible within the EHR and careful development and implementation to secure high reliability and minimise unintended consequences, including alert fatigue. The expert rule engine generating these alerts needs to be able to access data from multiple sources and execute complex algorithms. Data in the problem list, allergy list, other laboratory testing, the medication list, clinical encounter, etc., assure a better understanding of the individual case and improve accuracy of the CDS intervention. For example, in the case of warfarin and CYP2C9/VKORC1, the alert should trigger for initiation of therapy only, and the rule algorithm should be able to calculate a warfarin dose after considering multiple factors such as age, height/weight, indication, use of concomitant therapies that impact warfarin dosing, international normalised ratio level, and recent bleeding history. Moreover, the algorithm should be able to manage missing data elements and execute alternative notifications. Frequently, some of the data needed for PGx-CDS may not be accurately documented in the EHR or are inaccessible to the expert rule engine because the data are not documented in an appropriately structured format.” ([129], p. 257).

### 5.7. Evidence-Based Data on Effectiveness and Costs

To persuade physicians and health care organisations that the genomic data are evidence-based and cost efficient the clinical and economic utility of the data need to be validated by meticulously conducted RCTs at low risk of bias and sufficiently large to have power to measure the key outcomes. They need to include diverse patients and settings so that the cost benefits throughout the health care system can be estimated. This involves computing the adverse effects to patients and the costs of treating adverse drug reactions, hospitalisation, and legal liability awards to patients [129]. The IGNITE working group on economics has a process data to apply for codes to justify use and reimbursement [133]. Appropriately powered RCTs to enable assessment of the effects of multiple variant alleles on patients with multiple medications are needed.

## 6. Discussion

Individuals 65 and older with multiple illnesses usually have polypharmacy (≥5 medications and some ≥10) and although primary care physicians prescribe 70% of patients’ medications and renew specialists’ prescriptions their electronic medical records usually provide limited advice about prescribing multiple medications and avoiding ADRs.

The problem is due to the fact that medications are metabolised by the P450 CYP supergene family of 57 genes, of which twelve genes with 176 alleles are responsible for 80% of Phase 1 drug metabolism and 65–70% of Phase 2 drug clearance [77]. Prescribing needs to focus on the nine P450 genes which perform most metabolism. Humans have been using medications for only a few decades and thus these enzymes have not been subject to evolutionary pressure, resulting in great variation in P450 enzymatic capacity between individuals and large differences also by ancestry. Each patient receives one allele from the mother and one from the father and thus in each isoform patients may have two non-functioning alleles (“poor metabolisers”), one functioning and one non-functioning allele (“intermediate metabolisers”), two normally functioning (“normal metabolisers”), two faster functioning alleles (“rapid metabolisers”) or be ultrarapid metabolisers or even have multiple copies of genes. Thus, it is important for physicians to know how to prescribe safely for individuals with potential drug–drug and drug–drug–gene interactions and whose metabolism of medications varies substantially according to other medications they are taking and also their ancestry.

Currently, detailed pharmacogenomic advice is only available in some specialist clinics and services in major hospitals. However, this article provides detailed pharmacogenomic advice for primary care physicians and other physicians and this advice is equally useful for physicians in rural and remote areas worldwide.

### Limitations

For a substantial number of medications, it is unknown which P450 cytochrome isoform metabolises them or if a P450 isoform is involved. Also, DrugBank for medications not metabolised by P450 cytochrome isoforms reports as many as six complex pathways, each listing from four to sixty enzymes. These are not actionable by prescribing physicians other than prudently obtaining blood levels of medications or the results of using medications (e.g., levels of electrolytes such as potassium and sodium).

## 7. Conclusions

Two key databases provide data about the pathways humans use to metabolise medications. DrugBank is a detailed biochemical database of >13,000 medications and identifies those metabolised by individual or multiple P450 cytochrome isoforms and also those metabolised by complex pathways such as the solute carriers for diuretics. The Flockhart Tables also provide detailed lists of medications metabolised by P450 cytochrome isoforms and importantly lists all medications which are also inhibitors of P450 isoforms (resulting in blood levels of other medications which are high and thus overtreatment or ADRs or inducers of P450 isoforms (resulting in rapid metabolism, decrease in blood levels and undertreatment). An example of the importance of inducers is the prodrug codeine which is metabolised to morphine and patients who are inappropriate metabolisers will build up high and toxic levels.

Consulting the DrugBank or Flockhart tables for individual medications is not feasible during the workflow of primary care physicians Two key sources of detailed prescribing advice about P450 cytochrome metabolism based on the above tables are the Dutch Pharmacogenetics Working Group and the Clinical Pharmacogenetics Implementation Consortium. These databases provide detailed prescribing advice for each type of patient metabolism ranging from null metabolisers to ultrarapid metabolisers (Appendix A).

The genetics of the P450 cytochrome system of individual patients affects drug–drug and drug–drug–gene interactions which can result in dramatic increases in the blood levels of certain drugs. Detailed advice is provided in this article how to avoid multiple drug–drug–gene interactions (Appendix A).

The metabolism of each individual medication has to be identified by the physician. The importance of knowing how P450 enzymes and their variations differ between individuals and by ancestry can be illustrated by several important examples. Warfarin is metabolised by the CYP2C9 and CYP2D6 isoforms and its effects are also determined by the gene VKORC1 which affects Vitamin K metabolism. Because there is a narrow therapeutic range for warfarin, with INR levels <2 resulting in underprotection and levels >6 in substantial risk of bleeding and there are marked variations in enzymatic function between individuals there is strong advice to use a validated published pharmacogenetic algorithm that also incorporates patient data such as weight, age and comorbidities. Some drugs are pro-drugs such as codeine and enzymatic metabolism modifies them into another drug (morphine). In ultra-metabolisers toxic and fatal levels of morphine can occur but poor metabolisers will report insufficient pain relief.

Psychiatric medications are another group of medications that need close attention. Two thirds of all antidepressants are metabolised by two P450 isoforms, CYP2C19 and CYP2D6. The metabolism of antidepressants can vary by 42–50% due to genetic variations in individual patients, resulting usually in sub-therapeutic levels and failure to improve depression. Substantial variations in CYP2D6 function in individual patients may be caused by fact that CYP2D6 is the most highly polymorphic of the isoforms with substantial variability due to loss-of-function or gain of function in its 100+ alleles.

The care of patients with breast cancer illustrates the importance of verifying if a medication is ineffective due to patient genetic variation. More than 50% of breast cancer patients are estrogen receptor-positive and benefit by receiving tamoxifen which alters the transcription of estrogen-dependent genes and reduces relapse rates of breast cancer by 40%. However, for poor metabolisers of P450 CYP2D6 their levels of the active metabolites of tamoxifen are lower, their disease-free period is shorter, and it is important measure their tamoxifen levels and consider changing their medication to toremifene.

Already, Next Generation Sequencing datasets that use long reads are enabling identification of multiple alleles and variants. A review identified > 3.7 million single nucleotide variants (SNVs) and >350,000 insertions and deletions across ethnically different populations. Genes involved in drug absorption, distribution, metabolism, and excretion (ADME) are genetically diverse and complex and in 208 ADME genes, >69,000 SNVs have been identified. Although many are very rare, few are yet to be directly associated with diseases [134].

It is important for primary care physicians to use pharmacogenomic DSTs to prescribe correctly with this degree of variability in the human genome and with great variation by ancestry group. 

## Figures and Tables

**Figure 1 jpm-10-00084-f001:**
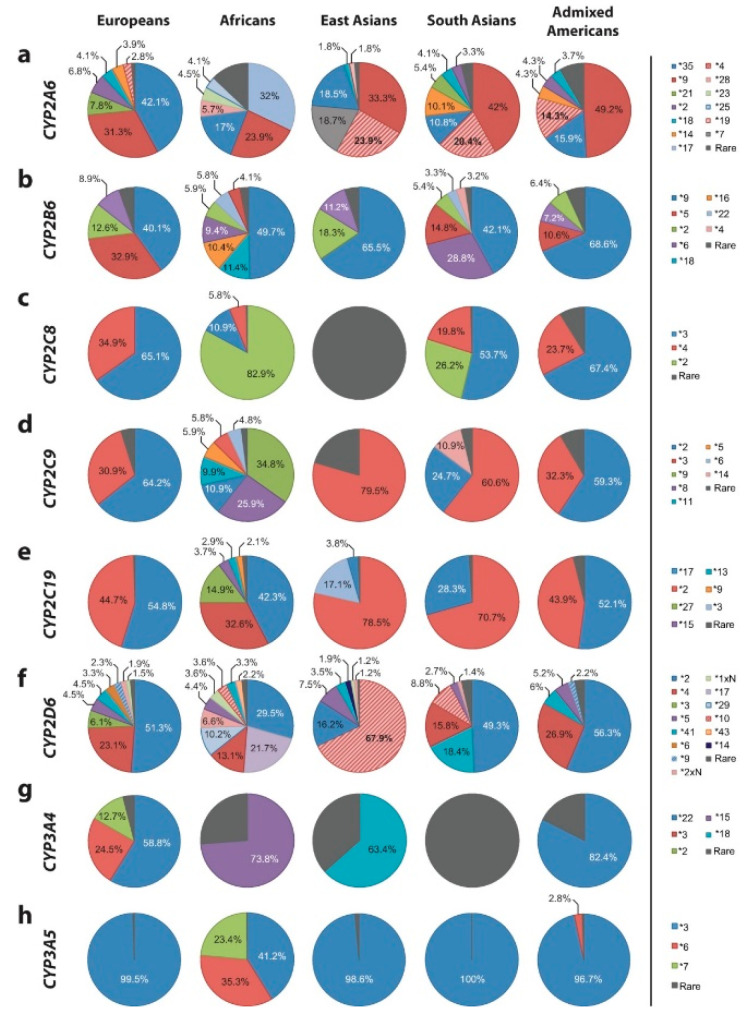
Worldwide Distribution of Cytochrome P450 Alleles: A Meta-analysis of Population-scale Sequencing Projects [77].

**Table 1 jpm-10-00084-t001:** Genotypes of CYP2C19 and CYP2D6 and effects on enzymatic capacity [78].

A. Genotypes and Enzymatic Activity of CYP2C19 and CYP2D6			
Genotype Functional	Diplotype	Categorisation ^1^	Enzymatic Capacity ^1^
**CYP2C19**			
CYP2C19 Null/Null	PM/PM	Poor	0%
CYP2C19Null/Wt	PM/NM	Intermediate	50%
CYP2C19Null/*17	PM/UM	Intermediate	60%
CYP2C19Wt/Wt	NM/NM	Normal	100%
CYP2C19Wt/*17	NM/UM	Ultrarapid	110%
CYP2C19*17/*17	UM/UM	Ultrarapid	120%
**CYP2D6**
CYP2D6Null/Null	PM/PM	Poor	0%
CYP2D6Null/*41	PM/IM	Intermediate	5%
CYP2D6Null/*9-10	PM/IM	Intermediate	15%
CYP2D6*41/*9-10	IM/IM	Intermediate OR Normal	20%
CYP2D6*9-10/*9-10	IM/IM	Intermediate OR Normal	30%
CYP2D6Wt/Null	NM/PM	Intermediate OR Normal	50%
CYP2D6Wt/*41	NM/IM	Normal	55%
CYP2D6Wt/*9-10	NM/IM	Normal	65%
CYP2D6Wt/Wt	NM/NM	Normal	100%
CYP2D6WtX3	UM/UM	Ultrarapid	150%

Source: [78] van Westrhenen, R.; Aitchison, K.J.; Ingelman-Sundberg, M.; Jukic’. M.M. Pharmacogenomics of Antidepressant and Antipsychotic Treatment: How Far Have We Got and Where Are We Going? *Front. Psychiatry*
**2020,**
*11*:94. doi: 10.3389/fpsyt.2020.00094. Notes: ^1^* and similar notations denote alleles of enzymes. NM, normal metaboliser; PM, poor metaboliser; IM, intermediate metaboliser; UM, ultrarapid metaboliser. The definition of the IM phenotype for CYP2D6 differs between sources.

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
