# Peer review of "Optimising Seniors’ Metabolism of Medications and Avoiding Adverse Drug Events Using Data on How Metabolism by Their P450 Enzymes Varies with Ancestry and Drug–Drug and Drug–Drug–Gene Interactions"

_jpm, 2020, doi:10.3390/jpm10030084_

Round 1
Reviewer 1 Report
The paper "Optimizing Seniors metabolism of Medications and avoiding serious adverse events..." by Roger E. Thomas is a very detailed review paper about the challenge of management of pharmacotherapy ans side effects due to competing metabolization by the cytochrome p 450 isoenzymes. Thai review gives a very good overview about the facts that general practitioners and internists should know. These data cross our way through practical medicine every day. Nothing of this is really new and Part of our daily prescription practice every day, but is good to have these data condensed in a single review.
Author Response
Many thanks for your kind review.
Reviewer 2 Report
In this article, the author investigated the benefits for primary care physicians to use pharmacogenomic information, especially P450 isoforms, in drug prescription. 75% of all medications are metabolised by P450 cytochrome enzymes. The author first summarized data from two key databases DrugBank and Flockhart tables. These two databases covers a majority of medications and corresponding metabolic pathway, and also lists their impact on regulating P450 activities. Regarding the inconvenient implementation of DrugBank and Flockhard, the author provides another two key resources, which includes detailed prescribing advice for each type of P450 genotype. The author then discussed how pharmacogenomics improves response, adverse effect, relapse, and cost in mental disease, cardiovascular disease, cancer, and other diseases. This article studied an important clinical issue, and provides a comprehensive summary of current clinical standards on pharmacogenomics, with a focus on P450 enzymes. I have the following comments.
- In the section 5 of the Results, the author discussed the importance and current situation of EMR, and summarized several pharmacogenomic based decision tools. This is very useful resource for our readers. I would suggest the author to generate a summary table of current available tools by disease area, and list their required patient information, advantages, and limitations. This will significantly improve the impact of this work.
- The article focused on the isoforms of P450 enzymes. But some key information about the sequencing test is not discussed. For example, how much is one test? Is this cost cover by insurance? Is this accessible for rural patients? What is the percentage of patients currently have P450 isoform profile?
- The title of this article is “Optimizing seniors’ medication by drug-drug-gene interaction”. But throughout the paper, I did not see any focus on seniors’ medication. Drug-drug-gene interaction is also not the emphasis. These two would be the most interested topics to most researchers and physicians. Because we mainly to prevent severe adverse events which happened much more frequently in senior patients with multiple conditions.
- The table 1 and 2 can be moved to supplement material.
- The legend in Figure 1 is too small to read.
- There are extensive grammar errors. E.g. Line 45 “serious drug events”, not “drug events”. Line 82 “the tables provide are clinically focused on the …”. Line 172 “Next Generation Sequencing” not “New Generation Sequencing”. P24 line 82 “repeated visits” not “repeat visits”
- Some abbreviations never explained in the main text. For example, RCTs. I can guess it is randomized clinical trial based on context, but it is better to have the full name when it first appears.
Author Response
Reviewer |
Changes made |
In this article, the author investigated the benefits for primary care physicians to use pharmacogenomic information, especially P450 isoforms, in drug prescription. 75% of all medications are metabolised by P450 cytochrome enzymes. The author first summarized data from two key databases DrugBank and Flockhart tables. These two databases covers a majority of medications and corresponding metabolic pathway, and also lists their impact on regulating P450 activities. Regarding the inconvenient implementation of DrugBank and Flockhart, the author provides another two key resources, which includes detailed prescribing advice for each type of P450 genotype. The author then discussed how pharmacogenomics improves response, adverse effect, relapse, and cost in mental disease, cardiovascular disease, cancer, and other diseases. This article studied an important clinical issue, and provides a comprehensive summary of current clinical standards on pharmacogenomics, with a focus on P450 enzymes. I have the following comments. |
Many thanks for your review |
In the section 5 of the Results, the author discussed the importance and current situation of EMR, and summarized several pharmacogenomic based decision tools. This is very useful resource for our readers. I would suggest the author to generate a summary table of current available tools by disease area, and list their required patient information, advantages, and limitations. This will significantly improve the impact of this work. |
I performed a search of the literature in Medline but did not find any summary articles of costs and achievements. Therefore, I consulted a colleague who is advising the Alberta Precision Laboratories (owned by the Province of Alberta) on how to improve the number and standardisation of their pharmacogenomic testing. I have incorporated in the article the key information in the e-mail reply I received from him: “The total number of patients who have received pharmacogenetic testing worldwide has not been published, is known only by the testing companies and varies by country and local healthcare systems. The larger insurers in the US are increasingly testing for specific indications such as when patients fail to benefit from antidepressants. A project of the provincially-owned Alberta Precision Laboratories is to standardise these tests. It is cheaper to obtain whole genome tests than for individual genomes.” (e-mail 21 July 2020 from Chad Bousman, MPH, PhD, Associate Professor Departments of Medical Genetics, Psychiatry, and Physiology & Pharmacology, University of Calgary) |
The article focused on the isoforms of P450 enzymes. But some key information about the sequencing test is not discussed. For example, how much is one test? Is this cost cover by insurance? Is this accessible for rural patients? What is the percentage of patients currently have P450 isoform profile? |
The US National Human Genome Research Project is one of the key funders of genomic research worldwide. Therefore, I performed a search of their website but did not identify any reports on the overall costs and achievements of genomic research although there are reports on highly focused research projects. The next 5-year Research Plan will be published in October 2020. www.genome.gov (Accessed 23 July 2020) |
The title of this article is “Optimizing seniors’ medication by drug-drug-gene interaction”. But throughout the paper, I did not see any focus on seniors’ medication. Drug-drug-gene interaction is also not the emphasis. These two would be the most interested topics to most researchers and physicians. Because we mainly to prevent severe adverse events which happened much more frequently in senior patients with multiple conditions. |
You are correct. I did a Medline and Embase search for this article but did not report it in the article. After I had submitted the article I realised that I had not identified clearly enough which studies focused on patients ≥ 65. So I repeated the search of Medline on 4 July 2020 using the search term pharmacogenomics and identified 11,963 studies and 194,072 systematic reviews or meta-analyses of which 221 were on pharmacogenomics. Of these 23 were of individuals ≥ 65 years. For Embase there were 11,050 studies of pharmacogenomics and 349,392 systematic reviews or metanalyses of which 282 were on pharmacogenomics. Of these four were of individuals ≥ 65 years. Each systematic review and also the reviews on those on individuals ≥ 65 years was assessed for relevance and if authors did not provide a mean or median age, this was computed from all studies comprising each systematic review. Evidence is reported if available for those ≥ 65 or if the authors noted there were no differences in drug metabolism between those < 65 or ≥ 65 years, otherwise data from other relevant studies is presented until data are available specifically for those ≥ 65. This new search resulted in the addition of five more studies focused on those ≥ 65 years.
For studies of progress implementing pharmacogenomics in practice I did a Medline search on 9 July 2020. This identified 55,417 citations for electronic medical records and when combined with pharmacogenomics yielded 176 systematic reviews or meta-analyses. There were no summary studies of the overall costs for hardware, software and training or performance across large healthcare systems and most publications were during 2014-2017 which reported initial experiences.
I also did a search on the US National Genome Research Project to identify current costs and progress but found only reports on highly focused topics and that is reported in the article and one 2020 report of an iterative search technique to reduce the number of drug interactions but it is in a very preliminary stage.
When I received your review I then re-read all the articles I had reviewed in my article and added any data for those ≥ 65. If the authors had not reported an average age for their systematic review I computed average ages from the individual studies. |
The table 1 and 2 can be moved to supplement material |
Tables 1 and 2 have been renamed Tables S1 and S2 and moved to the Supplemental material |
The legend in Figure 1 is too small to read. |
This will be amended in the editing phase |
There are extensive grammar errors. E.g. Line 45 “serious drug events”, not “drug events”. |
Each instance of “drug events” has been changed to: “serious drug events” |
Line 82 “the tables provide are clinically focused on the |
“The tables provide are clinically focused” has been changed to: “The tables are clinically focused” |
Line 172 “Next Generation Sequencing” not “New Generation Sequencing”. |
“New Generation Sequencing” has been changed to: “Next Generation Sequencing” |
P24 line 82 “repeated visits” not “repeat visits” |
“repeat visits” has been changed to: “repeated visits” |
Some abbreviations never explained in the main text. For example, RCTs. I can guess it is randomized clinical trial based on context, but it is better to have the full name when it first appears. |
RCT has been changed to: randomised controlled trial (RCT) and the manuscript has been searched for other abbreviations |
Round 2
Reviewer 2 Report
My concerns have been fully addressed.
I made a typo in my previous comment in "serious drug events". It should be "serious adverse event" or "adverse drug event". Please correct this.
Author Response
I have changed serious drug event to adverse drug event at every instance in the Ms.
This manuscript is a resubmission of an earlier submission. The following is a list of the peer review reports and author responses from that submission.
Round 1
Reviewer 1 Report
The article intended to provide the information/tools critical for physicians to make decisions when prescribing drugs for the seniors, who are likely to take many drugs concomitantly.
Review comments:
- The article would have been fitted for "Review" type manuscript, rather than "Article". Thus, the Method section is not necessary if the author switches the submission to the "Review" track. Or, the Method section should be written to provide literally the information relevant to research methodology, rather than summary of findings.
- The intrinsic objective of the study should be clearly described in “Introduction” section.
Overall, this Article is significant in providing the valuable information regarding drug-drug, drug-drug-gene interactions for the primary care physicians.
Reviewer 2 Report
The manuscript describes the variation of P450 enzyme activity in patients highly influencing blood levels and dosing. The author summarised recommendations of the Clinical Pharmacogenetics Implementation Consortium and the Dutch Pharmacogenetics Working Group. Although the topic is interesting there are major issues which the authors should address prior publication.
- This manuscript is a review and not an article. The author should therefore rewrite the manuscript now deleting Materials and Methods and change it to a new subchapter e.g. “Pharmacogenetic guidelines”. Rename also the subchapter “Results”. Parts of it e.g. 3.1.1 “mental illness” should be used to better explain tables.
- Title: has to be changed as there is no info about “optimizing senior’s metabolism” – most of the drugs listed in the tables are also prescribed to young patients. Title is too long – it should be shortened e.g. “Optimizing therapy of clinically used drugs metabolized by P450 enzymes”
- Table 1: Delete SLCO1B and VKORC1 genes (Simvastatin and Warfarin) as these genes do not encode P450 enzymes. If the author includes polymorphic genes important for therapy it has to discussed in a new subchapter. However, he should then use a title including other genes.
- Table 2: Don’t split the drug names
- Fig 1: should be deleted as it is hard to read.
- Subchapter 4.5.: line 403: statins are not metabolized by SLCO1B1 - as mentioned above either delete “other genes” like SLCO1B 1VKORC1 or better explain importance for drug therapy in a new subchapter.
- Author contributions: should be changed to a review manuscript
Round 2
Reviewer 2 Report
The authors did not substantially correct the mansucript. The manuscript is a review and not an article as it summarising literature data from Pubmed and recommendations of the Clinical Pharmacogenetics Implementation Consortium and the Dutch Pharmacogenetics Working Group. Although the topic is interesting there are still major issues which the authors should address prior publication.
- The author should therefore rewrite the manuscript changing it from an article to a review paper now deleting Materials and Methods and the heading "Results".
- Title: is still to long and should be shortened
- Table 1: Delete SLCO1B and VKORC1 genes (Simvastatin and Warfarin) as these genes do not encode P450 enzymes. If the author includes polymorphic genes important for therapy it has to discussed in a new subchapter. However, he should then use a title including other genes.
- Figure 1. should be deleated as it is hard to read
- Subchapter 4.5.: line 461: statins are not metabolized by SLCO1B1 - as mentioned above either delete “other genes” like SLCO1B 1VKORC1 or better explain importance for drug therapy in a new subchapter.
- Author contributions: should be changed to a review manuscript
- Tornio et al (lines 456-459, Floyed et al (lines491-494) Wen at al (lines 599-602)as well as a www. adress (line 636 are listed in the main body of the text and not in the list of references.